# Improving the accuracy of medical diagnosis with causal machine learning

Jonathan G. Richens [1✉], Ciarán M. Lee[1,2] & Saurabh Johri [1]

Machine learning promises to revolutionize clinical decision making and diagnosis. In medical diagnosis a doctor aims to explain a patient's symptoms by determining the diseases causing them. However, existing machine learning approaches to diagnosis are purely associative, identifying diseases that are strongly correlated with a patients symptoms. We show that this inability to disentangle correlation from causation can result in sub-optimal or dangerous diagnoses. To overcome this, we reformulate diagnosis as a counterfactual inference task and derive counterfactual diagnostic algorithms. We compare our counterfactual algorithms to the standard associative algorithm and 44 doctors using a test set of clinical vignettes. While the associative algorithm achieves an accuracy placing in the top 48% of doctors in our cohort, our counterfactual algorithm places in the top 25% of doctors, achieving expert clinical accuracy. Our results show that causal reasoning is a vital missing ingredient for applying machine learning to medical diagnosis.

[1] Babylon Health, 60 Sloane Ave, Chelsea, London SW3 3DD, UK. [2] University College London, Gower St, Bloomsbury, London WC1E 6BT, UK. ✉email: jonathan.richens@babylonhealth.com

Providing accurate and accessible diagnoses is a fundamental challenge for global healthcare systems. In the US alone an estimated 5% of outpatients receive the wrong diagnosis every year[1,2]. These errors are particularly common when diagnosing patients with serious medical conditions, with an estimated 20% of these patients being misdiagnosed at the level of primary care[3] and one in three of these misdiagnoses resulting in serious patient harm[1,4].

In recent years, artificial intelligence and machine learning have emerged as powerful tools for solving complex problems in diverse domains[5–7]. In particular, machine learning assisted diagnosis promises to revolutionise healthcare by leveraging abundant patient data to provide precise and personalised diagnoses[8–16]. Despite significant research efforts and renewed commercial interest, diagnostic algorithms have struggled to achieve the accuracy of doctors in differential diagnosis[17–23], where there are multiple possible causes of a patients symptoms.

This raises the question, why do existing approaches struggle with differential diagnosis? All existing diagnostic algorithms, including Bayesian model-based and Deep Learning approaches, rely on associative inference—they identify diseases based on how correlated they are with a patients symptoms and medical history. This is in contrast to how doctors perform diagnosis, selecting the diseases which offer the best causal explanations for the patients symptoms. As noted by Pearl, associative inference is the simplest in a hierarchy of possible inference schemes[24–26]. Counterfactual inference sits at the top of this hierarchy, and allows one to ascribe causal explanations to data. Here, we argue that diagnosis is fundamentally a counterfactual inference task. We show that failure to disentangle correlation from causation places strong constraints on the accuracy of associative diagnostic algorithms, sometimes resulting in sub-optimal or dangerous diagnoses. To resolve this, we present a causal definition of diagnosis that is closer to the decision making of clinicians, and derive counterfactual diagnostic algorithms to validate this approach.

We compare the accuracy of our counterfactual algorithms to a state-of-the-art associative diagnostic algorithm and a cohort of 44 doctors, using a test set of 1671 clinical vignettes. In our experiments, the doctors achieve an average diagnostic accuracy of 71.40%, while the associative algorithm achieves a similar accuracy of 72.52%, placing in the top 48% of doctors in our cohort. However, our counterfactual algorithm achieves an average accuracy of 77.26%, placing in the top 25% of the cohort and achieving expert clinical accuracy. These improvements are particularly pronounced for rare diseases, where diagnostic errors are more common and often more serious, with the counterfactual algorithm providing a better diagnosis for 29.2% of rare and 32.9% of very-rare diseases compared to the associative algorithm.

Importantly, the counterfactual algorithm achieves these improvements using the same disease model as the associative algorithm—only the method for querying the model has changed. This backwards compatibility is particularly important as disease models require significant resources to learn[20]. Our algorithms can thus be applied as an immediate upgrade to existing Bayesian diagnostic models, even those outside of medicine[27–30].

**Associative diagnosis**. Here, we outline the basic principles and assumptions underlying the current approach to algorithmic diagnosis. We then detail scenarios where this approach breaks down due to causal confounding, and propose a set of principles for designing diagnostic algorithms that overcome these pitfalls. Finally, we use these principles to propose two diagnostic algorithms based on the notions of necessary and sufficient causation.

Since its formal definition[31], model-based diagnosis has been synonymous with the task of using a model $\theta$ to estimate the likelihood of a fault component $D$ given findings $\mathcal{E}$[32],

$$P(D|\mathcal{E};\ \theta). \tag{1}$$

In medical diagnosis $D$ represents a disease or diseases, and findings $\mathcal{E}$ can include symptoms, tests outcomes and relevant medical history. In the case of diagnosing over multiple possible diseases, e.g., in a differential diagnosis, potential diseases are ranked in terms of their posterior. Model-based diagnostic algorithms are either discriminative, directly modelling the conditional distribution of diseases $D$ given input features $\mathcal{E}$ (1), or generative, modelling the prior distribution of diseases and findings and using Bayes rule to estimate the posterior,

$$P(D|\mathcal{E};\ \theta) = \frac{P(\mathcal{E}|D;\ \theta)P(D;\ \theta)}{P(\mathcal{E};\ \theta)}. \tag{2}$$

Examples of discriminative diagnostic models include neural network and deep learning models[8,10,15,33,34], whereas generative models are typically Bayesian networks[18,19,21,22,27,35,36].

How does this approach compare to how doctors perform diagnosis? It has long been argued that diagnosis is the process of finding causal explanations for a patient's symptoms[37–47]. For example[37], concludes "The generation of hypotheses is by habitual abduction. The physician relies on her knowledge of possible causes that explain the symptoms". Likewise[48] defines diagnosis as "the investigation or analysis of the cause or nature of a condition, situation, or problem". That is, given the evidence presented by the patient, a doctor attempts to determine the diseases that are the best explanation—the most likely underlying cause—of the symptoms presented. We propose the following causal definition of diagnosis,

*The identification of the diseases that are most likely to be causing the patient's symptoms, given their medical history.*

Despite the wealth of literature placing causal reasoning at the centre of diagnosis, to the best of our knowledge there are no existing approaches to model-based diagnosis that employ modern causal analysis techniques[49,50].

It is well known that using the posterior to identify causal relations can lead to spurious conclusions in all but the simplest causal scenarios—a phenomenon known as confounding[51]. For example, Fig. 1a shows a disease $D$ which is a direct cause of a symptom $S$. In this scenario, $D$ is a plausible explanation for $S$,

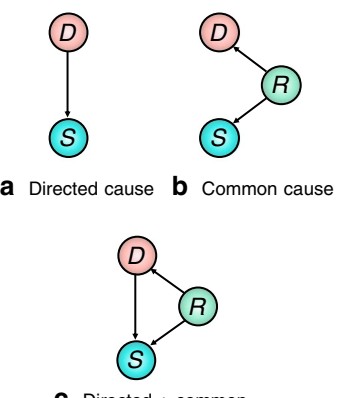

**a** Directed cause   **b** Common cause

**c** Directed + common

**Fig. 1 Three distinct causal structures for diseases and symptoms.**
**a** Disease $D$ is a direct cause of symptom $S$, **b** $D$ does not cause symptom $S$, but they are correlated by a latent common cause $R$, **c** $D$ is a direct cause of $S$ and a latent common cause $R$ is present.

and treating $D$ could alleviate symptom $S$. In Fig. 1b, variable $R$ is a confounder for $D$ and $S$, for example $R$ could be a genetic factor which increases a patients chance of developing disease $D$ and experiencing symptom $S$. Although $D$ and $S$ can be strongly correlated in this scenario, $P(D = T|S = T) \gg P(D = T)$ (where $D = T$ denotes the presence of $D$), $D$ cannot have caused symptom $S$ and so would not constitute a reasonable diagnosis. In general, diseases are related to symptoms by both directed and common causes that cannot be simply disentangled, as shown in Fig. 1c). The posterior (1) does not differentiate between these different scenarios and so is insufficient for assigning a diagnosis to a patient's symptoms in all but the simplest of cases, and especially when there are multiple possible causes for a patient's symptoms.

Example 1: An elderly smoker reports chest pain, nausea, and fatigue. A good doctor will present a diagnosis that is both likely and relevant given the evidence (such as angina). Although this patient belongs to a population with a high prevalence of emphysema, this disease is unlikely to have caused the symptoms presented and should not be put forward as a diagnosis. Emphysema is positively correlated with the patient's symptoms, but this is primarily due to common causes[52].

Example 2: Ref. [53] found that asthmatic patients who were admitted to hospital for pneumonia were more aggressively treated for the infection, lowering the sub-population mortality rate. An associative model trained on this data to diagnose pneumonia will learn that asthma is a protective risk factor—a dangerous conclusion that could result in a less aggressive treatment regime being proposed for asthmatics, despite the fact that asthma increases the risk of developing pneumonia. In this example, the confounding factor is the unobserved level of care received by the patient.

Real-world examples of confounding, such as Examples 1 and 2, have lead to increasing calls for causal knowledge to be properly incorporated into decision support algorithms in healthcare[54].

## Results

**Principles for diagnostic reasoning**. An alternative approach to associative diagnosis is to reason about causal responsibility (or causal attribution)—the probability that the occurrence of the effect $S$ was in fact brought about by target cause $D$[55]. This requires a diagnostic measure $\mathcal{M}(D, \mathcal{E})$ for ranking the likelihood that a disease $D$ is causing a patient's symptoms given evidence $\mathcal{E}$. We propose the following three minimal desiderata that should be satisfied by any such diagnostic measure,

  i. The likelihood that a disease $D$ is causing a patient's symptoms should be proportional to the posterior likelihood of that disease $\mathcal{M}(D, \mathcal{E}) \propto P(D = T|\mathcal{E})$ (consistency),

  ii. A disease $D$ that cannot cause any of the patient's symptoms cannot constitute a diagnosis, $\mathcal{M}(D, \mathcal{E}) = 0$ (causality),

  iii. Diseases that explain a greater number of the patient's symptoms should be more likely (simplicity).

The justification for these desiderata is as follows. Desideratum i) states that the likelihood that a disease explains the patient's symptoms is proportional to the likelihood that the patient has the disease in the first place. Desideratum ii) states that if there is no causal mechanism whereby disease $D$ could have generated any of the patient's symptoms (directly or indirectly), then $D$ cannot constitute causal explanation of the symptoms and should be disregarded. Desideratum iii) incorporates the principle of Occam's razor—favouring simple diagnoses with few diseases that can explain many of the

symptoms presented. Note that the posterior only satisfies the first desiderata, violating the last two.

**Counterfactual diagnosis**. To quantify the likelihood that a disease is causing the patient's symptoms, we employ counterfactual inference[56–58]. Counterfactuals can test whether certain outcomes would have occurred had some precondition been different. Given evidence $\mathcal{E} = e$ we calculate the likelihood that we would have observed a different outcome $\mathcal{E} = e'$, counter to the fact $\mathcal{E} = e$, had some hypothetical intervention taken place. The counterfactual likelihood is written $P(\mathcal{E} = e' \mid \mathcal{E} = e, \mathrm{do}(X = x))$ where $\mathrm{do}(X = x)$ denotes the intervention that sets variable $X$ to the value $X = x$, as defined by Pearl's calculus of interventions[49] (see Supplementary Note 3 for formal definitions).

Counterfactuals provide us with the language to quantify how well a disease hypothesis $D = T$ explains symptom evidence $S = T$ by determining the likelihood that the symptom would not be present if we were to intervene and 'cure' the disease by setting $\mathrm{do}(D = F)$, given by the counterfactual probability $P(S = F \mid S = T, \mathrm{do}(D = F))$. If this probability is high, $D = T$ constitutes a good causal explanation of the symptom. Note that this probability refers to two contradictory states of $S$ and so cannot be represented as a standard posterior[49,59]. In Supplementary Note 3 we describe how these counterfactual probabilities are calculated.

Inspired by this example, we propose two counterfactual diagnostic measures, which we term the expected disablement and expected sufficiency. We show in Theorem 1 at the end of this section that both measures satisfy all three of our desiderata.

**Definition 1** (Expected disablement) The expected disablement of disease $D$ is the number of present symptoms that we would expect to switch off if we intervened to cure $D$,

$$\mathbb{E}_{\mathrm{dis}}(D, \mathcal{E}) := \sum_{\mathcal{S}'} |\mathcal{S}_+ \setminus \mathcal{S}'_+| P(\mathcal{S}'|\mathcal{E}, do(D = F)) \quad (3)$$

where $\mathcal{E}$ is the factual evidence and $\mathcal{S}_+$ is the set of factual positively evidenced symptoms. The summation is calculated over all possible counterfactual symptom evidence states $\mathcal{S}'$ and $\mathcal{S}'_+$ denotes the positively evidenced symptoms in the counterfactual symptom state. $do(D = F)$ denotes the counterfactual intervention setting $D \to F$. $|\mathcal{S}_+ \setminus \mathcal{S}'_+|$ denotes the cardinality of the set of symptoms that are present in the factual symptom evidence but are not present in the counterfactual symptom evidence.

The expected disablement derives from the notion of necessary cause[50], whereby $D$ is a necessary cause of $S$ if $S = T$ if and only if $D = T$. The expected disablement therefore captures how well disease $D$ alone can explain the patient's symptoms, as well as the likelihood that treating $D$ alone will alleviate the patient's symptoms.

**Definition 2** (expected sufficiency) The expected sufficiency of disease $D$ is the number of positively evidenced symptoms we would expect to persist if we intervene to switch off all other possible causes of the patient's symptoms,

$$\mathbb{E}_{\mathrm{suff}}(D, \mathcal{E}) := \sum_{\mathcal{S}'} |\mathcal{S}'_+| P(\mathcal{S}'|\mathcal{E}, do(\mathsf{Pa}(\mathcal{S}_+) \setminus D = F)) \quad (4)$$

where the summation is over all possible counterfactual symptom evidence states $\mathcal{S}'$ and $\mathcal{S}'_+$ denotes the positively evidenced symptoms in the counterfactual symptom state. $\mathsf{Pa}(\mathcal{S}_+) \setminus D$ denotes the set of all direct causes of the set of positively evidenced symptoms excluding disease $D$, and $do(\mathsf{Pa}(\mathcal{S}_+) \setminus D = F)$ denotes the counterfactual intervention setting all $\mathsf{Pa}(\mathcal{S}'_+ \setminus D) \to F$. $\mathcal{E}$ denotes the set of all factual evidence. $|\mathcal{S}'_+|$

denotes the cardinality of the set of present symptoms in the counterfactual symptom evidence.

The expected sufficiency derives from the notion of sufficient cause[50], whereby $D$ is a sufficient cause of $S$ if the presence of $D$ can cause subsequent occurrence of $S$ but, as $S$ can have multiple causes, the presence of $S$ does not imply the prior occurrence of $D$. Note the use of sufficiency here is in line with[60] and does not refer to the INUS conditions \cite{mackie1974cement}. Typically, diseases are sufficient causes of symptoms (see Supplementary Note 4 for further discussion). By performing counterfactual interventions to remove all possible causes of the symptoms (both diseases and exogenous influences), the only remaining cause is $D$ and so we isolate its effect as a sufficient cause in our model. If we cannot assume that a disease is a sufficient cause of its symptoms, the expected disablement should be used. See Supplementary Note 8 for comparison of the expected disablement and sufficiency to other counterfactual measures. See Supplementary Note 9 for comparisons of the expected disablement and sufficiency and the posterior in some simple diagnostic models.

**Theorem 1** (Diagnostic properties of expected disablement and expected sufficiency). *Expected disablement and expected sufficiency satisfy the three desiderata.*
The proof is provided in Supplementary Notes 5 and 7.

**Structural causal models for diagnosis**. We now introduce the statistical disease models we use to test the diagnostic measures outlined in the previous sections. We then derive simplified expressions for the expected disablement and sufficiency in these models.

The disease models we use in our experiments are Bayesian Networks (BNs) that model the relationships between hundreds of diseases, risk factors and symptoms. BNs are widely employed as diagnostic models as they are interpretable and explicitly encode causal relations between variables—a prerequisite for causal and counterfactual analysis[49]. These models typically represent diseases, symptoms and risk factors as binary nodes that are either on (true) or off (false). We denote true and false with the standard integer notation 1 and 0 respectively.

A BN is specified by a directed acyclic graph (DAG) and a joint probability distribution over all nodes which factorises with respect to the DAG structure. If there is a directed arrow from node $X$ to $Y$, then $X$ is said to be a parent of $Y$, and $Y$ to be a child of $X$. A node $Z$ is said to be an ancestor of $Y$ if there is a directed path from $Z$ to $Y$. A simple example BN is shown in Fig. 2a, which depicts a BN modelling diseases, symptoms, and risk factors (the causes of diseases).

BN disease models have a long history going back to the INTERNIST-1[18], Quick Medical Reference (QMR)[19,20], and

PATHFINDER[21,22] systems, with many of the original systems corresponding to noisy-OR networks with only disease and symptom nodes, known as BN2O networks[36]. Recently, three-layer BNs as depicted in Fig. 2a have replaced these two layer models[23]. These models make fewer independence assumptions and allow for disease risk factors to be included. While our results will be derived for these models, they can be simply extended to models with more or less complicated dependencies[19,61].

In the field of causal inference, BNs are replaced by the more fundamental Structural Causal Models (SCMs), also referred to as Functional Causal Models and Structural Equation Models[59,62]. SCMs are widely applied and studied, and their relation to other approaches, such as probabilistic graphical models and BNs, is well understood[49,63]. The key characteristic of SCMs is that they represent each variable as deterministic functions of their direct causes together with an unobserved exogenous 'noise' term, which itself represents all causes outside of our model. That the state of the noise term is unknown induces a probability distribution over observed variables. For each variable $Y$, with parents in the model $X$, there is a noise term $u_Y$, with unknown distribution $P(u_Y)$ such that $Y = f(x, u_Y)$ and $P(Y = y | X = x) = \sum_{u_Y : f(x, u_Y) = y} P(U_Y = u_Y)$.

By incorporating knowledge of the functional dependencies between variables, SCMs enable us to determine the response of variables to interventions (such as treatments). Note that counterfactuals cannot in general be identified from data alone, and require modelling assumptions such as knowledge of the underlying structural equations[58,64]. As we now show, existing diagnostic BNs such as BN2O networks[36] are naturally represented as SCMs.

**Noisy-OR twin diagnostic networks**. When constructing disease models it is common to make additional modelling assumptions beyond those implied by the DAG structure. The most widely used of these correspond to 'noisy-OR' models[19]. Noisy-OR models are routinely used for modelling in medicine, as they reflect basic intuitions about how diseases and symptoms are related[65,66]. In addition, they support efficient inference[67] and learning[68,69], and allow for large BNs to be described by a number of parameters that grows linearly with the size of the network[68,70]. Under the noisy-OR assumption, a parent $D_i$ activates its child $S$ (causing $S = 1$) if (i) the parent is on, $D_i = 1$, and (ii) the activation does not randomly fail. The probability of failure, conventionally denoted as $\lambda_{D_i, S}$, is independent from all other model parameters. The 'OR' component of the noisy-OR states that the child is activated if any of its parents successfully activate it. Concretely, the value $s$ of $S$ is the Boolean OR function $\vee$ of its parents activation functions, $s = \vee_i f(d_i, u_i)$, where the activation functions take the form $f(d_i, u_i) = d_i \wedge \bar{u}_i$, $\wedge$ denotes the Boolean AND function, $d_i \in \{0, 1\}$ is the state of a given parent $D_i$ and $u_i \in \{0, 1\}$ is a latent noise variable ($\bar{u}_i := 1 - u_i$) with a probability of failure $P(u_i = 1) = \lambda_{D_i, S}$. The noisy-OR model is depicted in Fig. 1b. Intuitively, the noisy-OR model captures the case where a symptom only requires a single activation to switch it on, and 'switching on' a disease will never 'switch off' a symptom. For further details on noisy-OR disease modelling see Supplementary Note 2.

We now derive expressions for the expected disablement and expected sufficiency for these models using twin-networks method for computing counterfactuals introduced in[64,71]. This method represents real and counterfactual variables together in a single SCM—the twin network—from which counterfactual probabilities can be computed using standard inference techniques. This approach greatly amortizes the inference cost of calculating counterfactuals compared to abduction[49,72], which is

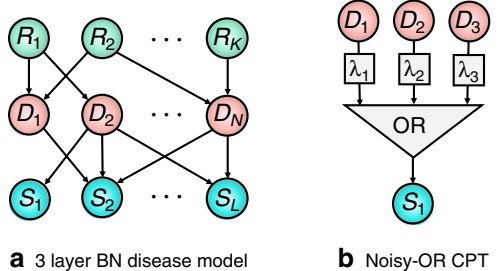

**a** 3 layer BN disease model   **b** Noisy-OR CPT

**Fig. 2 Generative structure of our diagnostic Bayesian networks. a** Three-layer Bayesian network representing risk factors $R_i$, diseases $D_j$ and symptoms $S_k$. **b** noisy-OR CPT. $S$ is the Boolean OR function of its parents, each with an independent probability $\lambda_i$ of being ignored, removing them from the OR function.

intractable for large SCMs. We refer to these diagnostic models as twin diagnostic networks, see Supplementary Note 3 for further details.

**Theorem 2** *For 3-layer noisy-OR BNs (formally described in Supplementary Notes 2 and 3, the expected sufficiency and expected disablement of disease $D_k$ are given by*

$$\frac{\sum_{\mathcal{Z} \subseteq \mathcal{S}_+} (-1)^{|\mathcal{Z}|} P(\mathcal{S}_- = 0, \mathcal{Z} = 0, D_k = 1 | \mathcal{R}) \tau(k, \mathcal{Z})}{P(\mathcal{S}_\pm | \mathcal{R})}, \quad (5)$$

*where for the expected sufficiency*

$$\tau(k, \mathcal{Z}) = \sum_{S \in \mathcal{S}_+ \setminus \mathcal{Z}} (1 - \lambda_{D_k, S}), \quad (6)$$

*and for the expected disablement*

$$\tau(k, \mathcal{Z}) = \sum_{S \in \mathcal{Z}} \left( 1 - \frac{1}{\lambda_{D_k, S}} \right), \quad (7)$$

*where $\mathcal{S}_\pm$ denotes the positive and negative symptom evidence, $\mathcal{R}$ denotes the risk-factor evidence, and $\lambda_{D_k, S}$ is the noise parameter for $D_k$ and $S$.*

The proof is provided by Theorem 2 in Supplementary Note 4 and by Theorem 4 in Supplementary Note 6.

**Experiments**. Here we outline our experiments comparing the expected disablement and sufficiency to posterior inference using the models outlined in the previous section. We introduce our test set which includes a set of clinical vignettes and a cohort of doctors. We then evaluate our algorithms across several diagnostic tasks.

**Diagnostic model and datasets**. One approach to validating diagnostic algorithms is to use electronic health records (EHRs)[8–12]. A key limitation of this approach is the difficulty in defining the ground truth diagnosis, where diagnostic errors result in mislabeled data. This problem is particularly pronounced for differential diagnoses because of the large number of candidate diseases and hence diagnostic labels, incomplete or inaccurate recording of case data, high diagnostic uncertainty and ambiguity, and biases such as the training and experience of the clinician who performed the diagnosis.

To resolve these issues, a standard method for assessing doctors is through the examination of simulated diagnostic cases or clinical vignettes[73]. A clinical vignette simulates a typical patient's presentation of a disease, containing a non-exhaustive list of evidence including symptoms, medical history, and basic demographic information such as age and birth gender[23]. This approach is often more robust to errors and biases than real data sets such as EHRs, as the task of simulating a disease given its known properties is simpler than performing a differential diagnosis, and has been found to be effective for evaluating human doctors[73–76] and comparing the accuracy of doctors to symptom checker algorithms[17,23,77,78].

We use a test set of 1671 clinical vignettes, generated by a separate panel of doctors qualified at least to the level of general practitioner (equivalent to board certified primary care physicians). The vignettes are generated independently of the assumptions underlying our disease model. Where possible, symptoms and risk factors match those in our disease model. However, to avoid biasing our study the vignettes include any additional clinical information as case notes, which are available to the doctors in our experiments. Each vignette is authored by a single doctor and then verified by multiple doctors to ensure that it represents a realistic diagnostic case. See Supplementary Note 10 for an example vignette. For each vignette the true disease is masked and the algorithm returns a diagnosis in the

form of a full ranking of all modelled diseases using the vignette evidence. The disease ranking is computed using the posterior for the associative algorithm, and the expected disablement or expected sufficiency for the counterfactual algorithms. Doctors provide an independent differential diagnosis in the form of a partially ranked list of candidate diseases.

In all experiments the counterfactual and associative algorithms use identical disease models to ensure that any difference in diagnostic accuracy is due to the ranking query used. The disease model used is a three-layer noisy-OR diagnostic BN as described above and in Supplementary Note 2. The BN is parameterised by a team of doctors and epidemiologists[23,78]. The model is specified independently of the test set of vignettes. The prior probabilities of diseases and risk factors are obtained from epidemiological data, and conditional probabilities are obtained through elicitation from multiple independent medical sources and doctors. The expected disablement and expected sufficiency are calculated using Theorem 2.

**Counterfactual vs associative rankings**. Our first experiment compares the diagnostic accuracy of ranking diseases using the posterior (1), expected disablement and expected sufficiency (5). For each of the 1671 vignettes the top-$k$ ranked diseases are computed, with $k = 1, \dots 20$, and the top-$k$ accuracy is calculated as fraction of the 1671 diagnostic vignettes where the true disease is present in the $k$-top ranking. The results are presented in Fig. 3. The expected disablement and expected sufficiency give almost identical accuracies for all $k$ on our test set, and for the sake of clarity we present the results for the expected sufficiency alone. The reasons for the similarity of these two measures on our test set is discussed in Supplementary Note 9. A complete table of results is present in the Supplementary Table 1.

For $k = 1$, returning the top ranked disease, the counterfactual algorithm achieves a 2.5% higher accuracy than the associative algorithm. For $k > 1$ the performance of the two algorithms diverge, with the counterfactual algorithm giving a large reduction in the error rate over the associative algorithm. For $k$

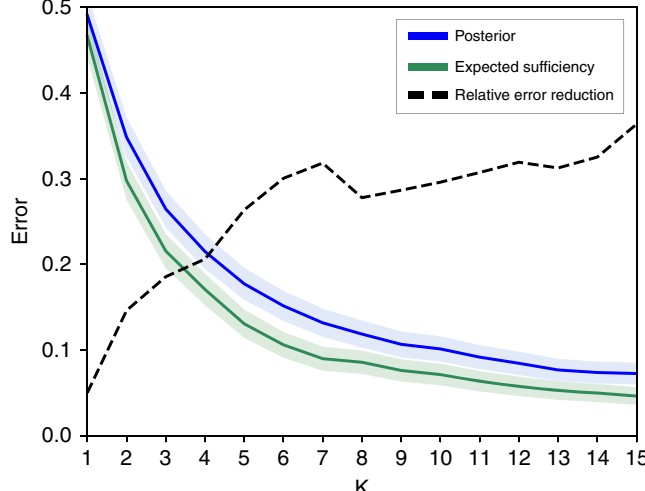

**Fig. 3 Top $k$ accuracy of Bayesian and counterfactual algorithms.** Figure shows the top $k$ error (1—accuracy) of the counterfactual (green line) and associative (blue line) algorithms over all 1671 vignettes vs $k$. Shaded regions give 95% confidence intervals. The black dashed line shows the relative reduction in error when switching from the associative to counterfactual algorithm, given by $1 - e_c / e_a$ where $e_a$ is the error rate of the associative algorithm, and $e_c$ is the error rate of the counterfactual algorithm. Results shown for $k = 1, \dots 15$, for complete results see the supplementary table 1.

**Table 1 Position of true disease in ranking stratified by rareness of disease.**

| | Vignettes | | | | | | |
| | All | VCommon | Common | Uncommon | Rare | VRare | ERare |
|---|---|---|---|---|---|---|---|
| N | 1671 | 131 | 413 | 546 | 353 | 210 | 18 |
| Mean position (A) | 3.81 | 2.85 | 2.71 | 3.72 | 4.35 | 5.45 | 4.22 |
| Mean position (C) | 3.16 | 2.5 | 2.32 | 3.01 | 3.72 | 4.38 | 3.56 |
| Wins (A) | 31 | 2 | 7 | 9 | 9 | 4 | 0 |
| Wins (C) | 412 | 20 | 80 | 135 | 103 | 69 | 5 |
| Draws | 1228 | 109 | 326 | 402 | 241 | 137 | 13 |

Table shows the mean position of the true disease for the associative (A) and counterfactual (C) algorithms. The results for expected disablement are almost identical to the expected sufficiency and are included in the Supplementary Notes. Results are stratified over the rareness of the disease (given the age and gender of the patient), where VCommon = Very common, VRare = very rare and ERare = extremely rare, and All is over all 1671 vignettes regardless of disease rarity. $N$ is the number of vignettes belonging to each rareness category. Mean(X) is the average position of the true disease for algorithm X. Wins (X) is the number of vignettes where algorithm X ranked the true disease higher than its counterpart, and Draws is the number of vignettes where the two algorithms ranked the true disease in the same position. For full results including uncertainties see the Supplementary Table 2.

> 5, the counterfactual algorithm reduces the number of misdiagnoses by ~30% compared to the associative algorithm. This suggests that the best candidate disease is reasonably well identified by the posterior, but the counterfactual ranking is significantly better at identifying the next most likely diseases. These secondary candidate diseases are especially important in differential diagnosis for the purposes of triage and determining optimal testing and treatment strategies.

A simple method for comparing two rankings is to compare the position of the true disease in the rankings. Across all 1671 vignettes we found that the counterfactual algorithm ranked the true disease higher than the associative algorithm in 24.7% of vignettes, and lower in only 1.9% of vignettes. On average the true disease is ranked in position $3.16 \pm 4.4$ by the counterfactual algorithm, a substantial improvement over $3.81 \pm 5.25$ for the associative algorithm (see Table 1).

In Table 1 we stratify the vignettes by the prior incidence rates of the true disease by very common, common, uncommon, rare and very rare. While the counterfactual algorithm achieves significant improvements over the associative algorithm for both common and rare diseases, the improvement is particularly large for rare and very-rare diseases, achieving a higher ranking for 29.2% and 32.9% of these vignettes respectively. This improvement is important as rare diseases are typically harder to diagnose and include many serious conditions where diagnostic errors have the greatest consequences.

**Comparing to doctors**. Our second experiment compares the counterfactual and associative algorithms to a cohort of 44 doctors. Each doctor is assigned a set of at least 50 vignettes (average 159), and returns an independent diagnosis for each vignette in the form of a partially ranked list of $k$ diseases, where the size of the list $k$ is chosen by the doctor on a case-by-case basis (average diagnosis size is 2.58 diseases). For a given doctor, and for each vignette diagnosed by the doctor, the associative and counterfactuals algorithms are supplied with the same evidence (excluding the free text case description) and each returns a top-$k$ diagnosis, where $k$ is the size of the diagnosis provided by the doctor. Matching the precision of the doctor for every vignette allows us to compare the accuracy of the doctor and the algorithms without constraining the doctors to give a fixed number of diseases for each diagnosis. This is important as doctors will naturally vary the size $k$ of their diagnosis to reflect their uncertainty in the diagnostic vignette.

The complete results for each of the 44 doctors, and for the posterior, expected disablement, and expected sufficiency ranking algorithms are included in the Supplementary Table 3. Figure 4 compares the accuracy of each doctor to the associative and

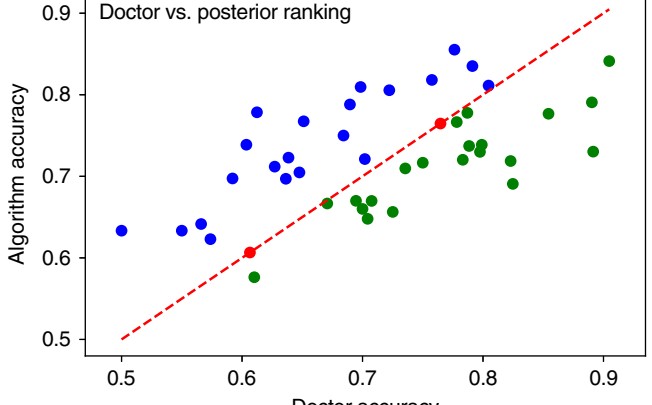

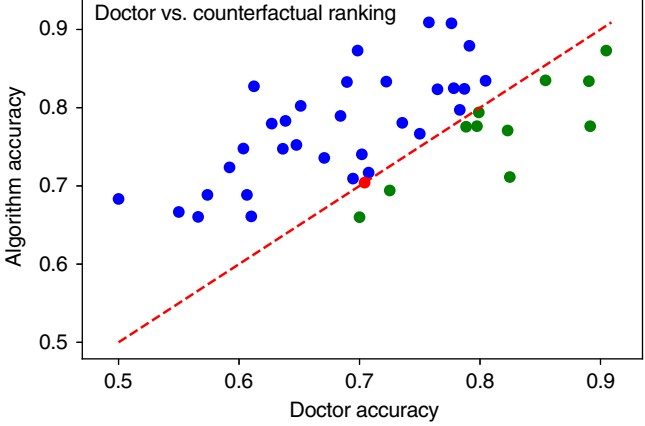

**Fig. 4 Mean accuracy of each doctor compared to Bayesian and counterfactual algorithms.** Figure shows the mean accuracy for each of the 44 doctors, compared to the posterior ranking (top) and expected sufficiency ranking (bottom) algorithms. The line $y = x$ gives a reference for comparing the accuracy of each doctor to the algorithm shadowing them. Points above the line correspond to doctors who achieved a lower accuracy than the algorithm (blue), points on the line are doctors that achieved the same accuracy as the algorithm (red), and below the line are doctors that achieved higher accuracy than the algorithm (green). The linear correlation can be explained by the variation in the difficulty of the sets of vignettes diagnosed by each doctor. Sets of easier/harder vignettes results in higher/lower doctor and algorithm accuracy scores. As the results for the expected disablement and expected sufficiency are almost identical, we show only the results for the expected sufficiency. Complete results are listed in the Supplementary Table 3. All figures generated using matplotlib version 3.2.1.

**Table 2 Group mean accuracy of doctors and algorithms.**

| Agent | Accuracy (%) | $N_{\geq D}$ | $N_{\geq A}$ | $N_{\geq C1}$ | $N_{\geq C2}$ |
|---|---|---|---|---|---|
| D | 71.40 ± 3.01 | – | 23 (8) | 12 (4) | 13 (5) |
| A | 72.52 ± 2.97 | 23 (9) | – | 1 (0) | 1 (0) |
| C1 | 77.26 ± 2.79 | 33 (20) | 44 (13) | – | 36 (0) |
| C2 | 77.22 ± 2.79 | 33 (19) | 44 (14) | 32 (0) | – |

The mean accuracy of the doctors D, associative A and counterfactual algorithms (C1 = expected sufficiency, C2 = expected disablement), averaged over all experiments. $N \geq K$ gives the number of trials (one for each doctor) where this agent achieved a mean accuracy the same or higher than the mean accuracy of agent $K \in \{D, A, C1, C2\}$. The bracketed term is the number of trials where the agent scored the same or higher accuracy than agent K to 95% confidence, determined by a one sided binomial test.

counterfactual algorithms. Each point gives the average accuracy for one of the 44 doctors, calculated as the proportion of vignettes diagnosed by the doctor where the true disease is included in the doctor's differential. This is plotted against the accuracy that the corresponding algorithm achieved when diagnosing the same vignettes and returning differentials of the same size as that doctor.

Doctors tend to achieve higher accuracies in case sets involving simpler vignettes—identified by high doctor and algorithm accuracies. Conversely, the algorithm tends to achieve higher accuracy than the doctors for more challenging vignettes—identified by low doctor and algorithm accuracies. This suggests that the diagnostic algorithms are complimentary to the doctors, with the algorithm performing better on vignettes where doctor error is more common and vice versa.

Overall, the associative algorithm performs on par with the average doctor, achieving a mean accuracy across all trails of 72.52 ± 2.97% vs 71.4 ± 3.01% for doctors. The algorithm scores higher than 21 of the doctors, draws with 2 of the doctors, and scores lower than 21 of the doctors. The counterfactual algorithm achieves a mean accuracy of 77.26 ± 2.79%, considerably higher than the average doctor and the associative algorithm, placing it in the top 25% of doctors in the cohort. The counterfactual algorithm scores higher than 32 of the doctors, draws with 1, and scores a lower accuracy than 12 (see Table 2).

In summary, we find that the counterfactual algorithm achieves a substantially higher diagnostic accuracy than the associative algorithm. We find the improvement is particularly pronounced for rare diseases. While the associative algorithm performs on par with the average doctor, the counterfactual algorithm places in the upper quartile of doctors.

## Discussion

Poor access to primary healthcare and errors in differential diagnoses represent a significant challenge to global healthcare systems[1–4,79,80]. If machine learning is to help overcome these challenges, it is important that we first understand how diagnosis is performed and clearly define the desired output of our algorithms. Existing approaches have conflated diagnosis with associative inference. While the former involves determining the underlying cause of a patient's symptoms, the latter involves learning correlations between patient data and disease occurrences, determining the most likely diseases in the population that the patient belongs to. While this approach is perhaps sufficient for simple causal scenarios involving single diseases, it places strong constraints on the accuracy of these algorithms when applied to differential diagnosis, where a clinician chooses from multiple competing disease hypotheses. Overcoming these constraints requires that we fundamentally rethink how we define diagnosis and how we design diagnostic algorithms.

We have argued that diagnosis is fundamentally a counterfactual inference task and presented a causal definition of diagnosis. We have derived two counterfactual diagnostic measures, expected disablement and expected sufficiency, and a class of diagnostic models—twin diagnostic networks—for calculating these measures. Using existing diagnostic models we have demonstrated that ranking disease hypotheses by these counterfactual measures greatly improves diagnostic accuracy compared to standard associative rankings. While the associative algorithm performed on par with the average doctor in our cohort, the counterfactual algorithm places in the top 25% of doctors in our cohort—achieving expert clinical accuracy. The improvement is particularly pronounced for rare and very-rare diseases, where diagnostic errors are typically more common and more serious, with the counterfactual algorithm ranking the true disease higher than the associative algorithm in 29.2% and 32.9% of these cases respectively. Importantly, this improvement comes 'for free', without requiring any alterations to the disease model. Because of this backward compatibility our algorithm can be used as an immediate upgrade for existing Bayesian diagnostic algorithms including those outside of the medical setting[27–30,81].

Whereas other approaches to improving clinical decision systems have focused on developing better model architectures or exploiting new sources of data, our results demonstrate a new path towards expert-level clinical decision systems—changing how we query our models to leverage causal knowledge. Our results add weight to the argument that machine learning methods that fail to incorporate causal reasoning will struggle to surpass the capabilities of human experts in certain domains[24].

Our results present the first evidence of the superiority of counterfactual over associative reasoning in a complex real-world task. The question of how to incorporate causal and counterfactual reasoning into other machine learning methods beyond structural causal models, for example in Deep Learning for image classification[82,83] and deep generative models[84–86], is an active area research. We hope that the results presented in our article will further motivate this area of research, by presenting a new application for improving diagnostic accuracy using counterfactual inference.

While we have focused on comparing our algorithms to doctors, future experiments could determine the effectiveness of these algorithms as clinical support systems—guiding doctors by providing a second opinion diagnosis. Given that our algorithm appears to be complimentary to human doctors, performing better on vignettes that doctors struggle to diagnose, it is likely that the combined diagnosis of doctor and algorithm will be more accurate than either alone.

## Methods

The proofs and further exposition of our disease models and inference methods are contained in Supplementary Information. In Supplementary Note A and B we address the preliminaries and framework within which we derive our proofs, introducing the framework of structural causal models, defining noisy-OR networks as structural causal models, and detailing their assumptions with respect to disease modelling. In Supplementary Note C we introduce counterfactual inference in structural causal models and the twin-networks inference scheme, and derive the twin networks used to compute the expected disablement and sufficiency. In Supplementary Note D we derived the expression for the expected sufficiency and in Supplementary Note E we prove that it satisfies our desiderata. In Supplementary Note F we derived the expression for the expected disablement and in Supplementary Note E we prove that it satisfies our desiderata. In Supplementary Note J we provide an example of the clinical vignettes used in our experiments. In Supplementary Note K we provide our full experimental results.

**Reporting summary**. Further information on research design is available in the Nature Research Reporting Summary linked to this article.

## Data availability

The data that support the findings of this study are available on https://github.com/babylonhealth/counterfactual-diagnosis. Any features of the vignettes not used to generate the methods or results of the study have been removed or de-identified prior to

sharing. Accredited researchers may request access to the complete clinical dataset for the purpose of checking the validity of the clinical vignettes used in the study by contacting the corresponding author. Access will be vetted by the Babylon Health access committee and will take place within the Babylon health intranet and under a non-disclosure agreement.

## Code availability
The code used to generate results shown in this study is available at https://github.com/babylonhealth/counterfactual-diagnosis[87].

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

## Acknowledgements

The authors would like to thank Kostis Gourgoulias, Chris Lucas, Yura Perov, Adam Baker and Albert Buchard for discussions. The authors would like to thank Peter Spirtes for valuable suggestions for the manuscript.

## Author contributions

All authors contributed to the original idea for the project. J.G.R. conceived the project and acted as project leader. J.G.R. and C.M.L. wrote the first draft of the paper and the revised versions with input from S.J. Proofs, experiments, code, tables, and figures contributed by J.G.R. All authors discussed the content and contributed to editing the manuscript.

## Competing interests

All the authors in the article are employees of Babylon Health.
