## [Peer Review File · Nature Communications]

Reviewers' Comments:

Reviewer #1:

Remarks to the Author:

I am thankful for the opportunity to review this. Before reading it, I had no strong prior opinion on diagnostic system besides some openness toward new approaches fed by my experiences with conceptual and practical problems of psychiatric diagnoses.

The writing of the manuscript is very appealing, and the theoretical foundation, definitions and concepts appear sound and convergent. Introducing causality into machine learning so that diseases could indeed be better diagnosed is just the kind of breakthrough that Judea Pearl desires generally for artificial intelligence. Therefore, the paper has maximum relevance and scientific merit for me. Frankly, I was not able to check all the mathematical details.

The method is tested only in a small sample, and its performance might well differ with other doctors and patients with other diseases. However, this is generally fine for introducing the method, and I am curious to see how it performs with different data. However, the readership needs to be ensured that the data presented were the only data the method was used with, and no method specification was done after inspecting the data.

One point that requires some revision is that the paper sometimes reads as if the method was able to reveal causal answers on its own. Consider the sentence: "Counterfactuals can test whether certain outcomes would have occurred had some precondition been different." This suggests that data and the method were together sufficient to answer this question. This is not the case since such testing also requires a model fed with substantive assumptions e.g. on common causes (or absence of measurement error) to compute the counterfactuals. The model behind the analysis, however, is displayed in figure 2, and the explanatory text mentions that the model could be modified. Anyway, the results may strongly depend on the specifications here and perform worse under changes that might better map reality. Particularly, such a model must be complete with regard to the shared factors, and this is certainly violated in the calculated example. This needs to become more transparent. Otherwise I fear that researchers overlook this between the mathematical notation and easily become over-confident in using the method.

I recommend the following two references for a general discussion:

Greenland, S. For and Against Methodologies: Some Perspectives on Recent Causal and Statistical Inference Debates. (2017). *European Journal of Epidemiology*, 32(1), 3-20.

Gigerenzer, G., & Marewski, J. N. (2015). Surrogate science: The idol of a universal method for scientific inference. *Journal of Management*, 41(2)

Reviewer #2:

None

Reviewer #3:

None

Reviewer #4:

Remarks to the Author:

The main purpose of the article is to propose a new causal goal of diagnosis – to identify the diseases most likely to be causing a patient's symptoms. This goal is fleshed out in two ways, the probability of sufficiency of a cause, and the probability of disablement (that turning off the cause would remove the symptoms). The authors provide new formulae for calculating the probability of

sufficiency and the probability of disablement, given a 3 tiered noisy-or Bayesian network symptom checkers. These formulae can be employed on existing symptom checkers (in place of the usual goal of finding the disease with the highest probability), and in experiments on a large number of vignettes, provide diagnoses more closely matching the true causes in the vignette.

The article is technically sound and generally well written. However, there are a number of questions that are not addressed in the paper (or referred to only briefly in the appendix) that are listed below. The subject is an important one, and the experiments reveal a marked improvement using the counterfactual rather than the associative reasoning. The topic is of theoretical interest, and possibly useful in practice.

There are a number of ways in which the exposition could be improved.

Generally speaking, the article says relatively little about how the proposed diagnostic definitions (expected disablement and expected sufficiency) differ from other previously proposed counterfactual definitions, such as effect of treatment on the treated (or e.g. probability of necessity and probability of sufficiency). It is pointed out in the appendix that the expected disablement is similar to the effect of treatment on the treated, but not in the main body of the paper, and it does not go into any detail.

In addition, while it is made clear in the appendix that the two quantities that are computed (expected sufficiency and expected disablement) are really only approximations to what is actually desired. ("The question of how we can define and quantify causal explanations in general models is an area of active research [45, 81-83] and the approach we propose here cannot be applied to all conceivable SCMs. For example, if you had a symptom that can be present only if two parents diseases D1 and D2 are both present, then neither of these parents in isolation is a sufficient cause (individually, $D1 = 1$ and $D2 = 1$ are necessary but not sufficient to cause $S = 1$).") There is also some justification of the choices of expected sufficiency and expected disablement for diagnosis that would be better included in the main body of the paper.

It would be very useful to the reader to have a simple (even toy example) where the associative inference is different than the expected sufficiency or expected disablement inference (and where the latter two are different from each other). This could give some intuition about why the latter two are more in line with our intuitions about the correct diagnosis than the former.

It also would be useful to the reader to have an example of the vignettes that were used in the experiment (possibly in the appendix) to see how much detail they provide, etc. It is also not clear whether the vignettes were designed to satisfy assumptions built into the symptom checker. For example, were symptoms that were only caused by multiple diseases (in contrast to what is assumed by the symptom checker) excluded? The paper says "In Appendix A.4 we present a different counterfactual query that captures causality in this case by reasoning about necessary treatments", but there is no Appendix A.4. How often would such diseases be expected to occur? Another thing that makes diagnosis difficult is having multiple diseases causing multiple symptoms. Do these kinds of cases occur in the vignettes?

The results of the experiment found that expected disablement and expected sufficiency performed very similarly. This is somewhat mysterious, since they are conceptually and computationally quite different, it is far from obvious why this should be the case. Do the authors have any insight into this result?

An interesting question raised by this article that the authors do not address is the tradeoff between using a 3-level noisy-or Bayesian network together with expected disablement or expected sufficiency, and some other non-counterfactual machine learning methods, which may improve the associative reasoning at the cost of not allowing for counterfactual reasoning. The 3-level noisy-or Bayesian network does not allow for latent variables, feedback, continuous variables, etc. It is not clear whether machine learning algorithms that do not have these limitations might

perform better simply due to improved associative reasoning, even if they can't do the counterfactual reasoning. It is not clear whether there are other associative systems that could be applied to the same vignettes, since there is not much information about what the vignettes covered.

The article states "Note that (5) recovers the standard posterior $P(D_k = 1|E)$ in the limit that $\tau(D_k, Z) \rightarrow 1$ for all Z ." I don't see how this could hold for all Z for the case of expected disablement, or for more than one present symptom in the case of expected sufficiency or at all for expected disablement.

In the author's (and others) terminology a disease is a sufficient cause for a symptom if no other disease is needed to produce the symptom. This allows for the possibility that the disease occurs but the symptom does not. There is another use of the term "sufficiency" that is common in discussions of causality which would not allow for the latter possibility (e.g. in discussions of INUS conditions for causation). The authors should include a footnote to avoid this confusion. This is clear in the appendices but not the main body of the paper.

Response to referees for NCOMMS-20-10366

Dear Dr. Righetto,

We are glad to know that the two reviews are very positive. We thank the reviewers for their comments and suggestions, which we address in the following letter.

Editor comments. In addition to the reviewer comments, the editor raises the following point, *“we would very much appreciate a more detailed but brief discussion on how your method might be applied to the field of artificial intelligence tools (eg Deep Learning) for medical imaging which you only (but widely) cite in your article.”*

Extending causal and counterfactual reasoning to the image domain has recently received much attention. Examples include applying causal methods to deconfound deep learning in medical image classification [1,2], and learning causal deep generative models for images [3-5]. In principle our core results—reframing diagnosis as a counterfactual inference task—could be applied to the medical imaging domain through these emerging methods. This goes beyond the scope of our article and our future research will follow this direction. We hope that the results presented in our article will further motivate research into causal and counterfactual methods in image classification by presenting a new application in improving medical diagnosis. We agree that it will improve the quality of the paper to acknowledge these areas of research, and have rewritten paragraph 4 of the discussion section to address these issues and cite the relevant existing papers.

Reviewer 1 describes the article as “the kind of breakthrough that Judea Pearl desires generally for artificial intelligence” and that it “has maximum relevance and scientific merit”. The reviewer then makes several suggestions which we address here.

Point 1. *“the readership needs to be ensured that the data presented were the only data the method was used with, and no method specification was done after inspecting the data.”*

The model was specified independently of the test data, and no post-selection was performed on the data. We have included clarifying remarks in section 4 A paragraph 4.

Point 2. *“One point that requires some revision is that the paper sometimes reads as if the method was able to reveal causal answers on its own. Consider*

the sentence: "Counterfactuals can test whether certain outcomes would have occurred had some precondition been different." This suggests that data and the method were together sufficient to answer this question. This is not the case since such testing also requires a model feeded with substantive assumptions e.g on common causes (or absence of measurement error) to compute the counterfactuals. The model behind the analysis, however, is displayed in figure 2, and the explanatory text mentions that the model could be modified. Anyway, the results may strongly depend on the specifications here and perform worse under changes that might better map reality. Particularly, such a model must be complete with regard to the shared factors, and this is certainly violated in the calculated example. This needs to become more transparent. Otherwise I fear that researchers overlook this between the mathematical notation and easily become over-confident in using the method. "

The reviewer points out that counterfactual inferences are valid only up to the underlying causal and functional modelling assumptions. We agree that this point should be clarified in the manuscript, and we have updated the previous discussion of this point in the final paragraph of section 3 A with additional clarifying remarks and citations to materials on counterfactual identifiability and modelling assumptions.

Point 3. *"I recommend the following two references for a general discussion: Greenland, S. For and Against Methodologies: Some Perspectives on Recent Causal and Statistical Inference Debates. (2017). European Journal of Epidemiology, 32(1), 3-20. Gigerenzer, G., Marewski, J. N. (2015). Surrogate science: The idol of a universal method for scientific inference. Journal of Management, 41(2)"*

References have been added.

Reviewer 4 describes the article as "technically sound", and that "The subject is an important one, and the experiments reveal a marked improvement using the counterfactual rather than the associative reasoning". The reviewer then suggests several improvements, which we address here.

Point 1. *"Generally speaking, the article says relatively little about how the proposed diagnostic definitions (expected disablement and expected sufficiency) differ from other previously proposed counterfactual definitions, such as effect of treatment on the treated (or e.g. probability of necessity and probability of sufficiency). It is pointed out in the appendix that the expected disablement is similar to the effect of treatment on the treated, but not in the main body of the paper, and it does not go into any detail."*

We have included a new appendix H, where we discuss the relation of our diagnostic definitions to the effect of treatment on the treated, probability of necessity, and probability of sufficiency, and have detailed why our proposed

diagnostic measures are better suited for the task of diagnosis. We refer to this appendix at the end of section 2, in the last sentence before theorem 1, in the main body of the manuscript.

Point 2. *“In addition, while it is made clear in the appendix that the two quantities that are computed (expected sufficiency and expected disablement) are really only approximations to what is actually desired. (“The question of how we can define and quantify causal explanations in general models is an area of active research [45, 81-83] and the approach we propose here cannot be applied to all conceivable SCMs. For example, if you had a symptom that can be present only if two parents diseases $D1$ and $D2$ are both present, then neither of these parents in isolation is a sufficient cause (individually, $D1 = 1$ and $D2 = 1$ are necessary but not sufficient to cause $S = 1$).”) There is also some justification of the choices of expected sufficiency and expected disablement for diagnosis that would be better included in the main body of the paper.”*

We have clarified our discussion of these points, including the hypothetical example of ‘necessary but insufficient diseases’ in the third paragraph of appendix D, and we have added a brief discussion of these points to the final paragraph of section 2 in the main body. We note that under the modelling assumption that diseases are sufficient causes of symptoms, which is a standard assumption in medical literature, the expected sufficiency is the desired quantity. The expected disablement is proposed as an alternative measure that does not assume sufficient causes, and is robust to the above example.

Point 3. *“It would be very useful to the reader to have a simple (even toy example) where the associative inference is different than the expected sufficiency or expected disablement inference (and where the latter two are different from each other). This could give some intuition about why the latter two are more in line with our intuitions about the correct diagnosis than the former.”*

We have included a new appendix I detailing examples of simple cases where the expected disablement and sufficiency give the correct diagnosis, whereas the associative measure (the posterior) results in a spurious diagnosis.

Point 4. *“It also would be useful to the reader to have an example of the vignettes that were used in the experiment (possibly in the appendix) to see how much detail they provide, etc.”*

An example vignette has been added to appendix J.

Point 5. *“It is also not clear whether the vignettes were designed to satisfy assumptions built into the symptom checker. For example, were symptoms that were only caused by multiple diseases (in contrast to what is assumed by the symptom checker) excluded? [...] How often would such diseases be expected to occur? Another thing that makes diagnosis difficult is having multiple diseases*

causing multiple symptoms. Do these kinds of cases occur in the vignettes? ”

Our vignettes were constructed independently of any of the assumptions (causal, functional or otherwise) that are present in the symptom checker. We have extended the statement in section 3 A paragraph 2 of the article to clarify this point.

The reviewer asks how common it is to find symptoms that require multiple diseases to be present. We give a toy example of this case in appendix D of the article, as an extreme example of (possible) causal interferences between diseases. We are aware of no symptoms that behave this way in reality. However, if such symptoms do exist they would not be excluded for our vignettes. Consider the case that a vignette models a disease D_1 , which is frequently accompanied by a symptom S. This symptom would be included by the clinicians in the vignette, even if the presence of this symptom requires the presence of a secondary disease D_2 .

Finally, the reviewer asks if vignettes modelling multiple diseases are used. In section 3 A paragraph 2 it is stated that the vignettes model realistic presentations of single diseases. This is common practice in the medical literature, and we note that vignettes modelling single diseases do not preclude multimorbidities. For example, a realistic presentation of a disease that frequently is associated with comorbidities will reflect this in the symptom profile.

Point 6. *“The paper says “In Appendix A.4 we present a different counterfactual query that captures causality in this case by reasoning about necessary treatments”, but there is no Appendix A.4.”*

Typo corrected.

Point 7. *“The results of the experiment found that expected disablement and expected sufficiency performed very similarly. This is somewhat mysterious, since they are conceptually and computationally quite different, it is far from obvious why this should be the case. Do the authors have any insight into this result?”*

We have investigated this result, and have presented an exploration of why this is the case in appendix I and a reference to this in the first paragraph of section 4 B. We have found that the ranking given by these two measures coincide in simple noisy-OR models when there are no prior correlations between diseases. Because the majority of diseases in our model only have weak prior correlations, we conclude that this is the most likely explanation for the similarity of these two measures on our test set. This discussion is included in appendix I and referenced in the first paragraph of section 4 B.

Point 8. *“An interesting question raised by this article that the authors do not address is the trade-off between using a 3-level noisy-or Bayesian network together with expected disablement or expected sufficiency, and some other non-counterfactual machine learning methods, which may improve the associative*

reasoning at the cost of not allowing for counterfactual reasoning. The 3-level noisy-or Bayesian network does not allow for latent variables, feedback, continuous variables, etc. It is not clear whether machine learning algorithms that do not have these limitations might perform better simply due to improved associative reasoning, even if they can't do the counterfactual reasoning. It is not clear whether there are other associative systems that could be applied to the same vignettes, since there is not much information about what the vignettes covered."

We agree with the reviewer that there may be more expressive associative methods that achieve a higher accuracy on our test set. However, in section 1 A (and appendix J) we show that there are realistic diagnostic scenarios where any purely associative model will return spurious diagnoses. Regardless of the model capacity, these methods will fit to data that is biased by confounding. Hence, any method that is fundamentally limited to associative reasoning must be sub-optimal for these cases, even compared to simple causal models. These cases appear to be quite common in primary diagnosis, as evidenced by our results. The question of a trade-off can then be replaced with another question: can more expressive machine learning methods be extended to incorporate counterfactual reasoning?

This question has received much attention lately, including extending causal and counterfactual inference to deep learning in image classification [1,2] and deep generative models [3-5]. This is an active area of research which is beyond the scope of this article. This notwithstanding, we agree that it will improve the quality of the paper to acknowledge these areas of research. We have rewritten paragraph 4 of the discussion section to address these issues and cite the relevant existing papers.

We also note that there are structural causal models involving latent variables (as ours does), continuous variables and feedback exist, and we expect these models to benefit from our approach. We have also included an example vignette in Appendix J at the reviewers recommendation.

Point 9. *"The article states "Note that (5) recovers the standard posterior $P(D_k = 1|\mathcal{E})$ in the limit that $\tau(D_k, Z) \rightarrow 1$ for all Z ." I don't see how this could hold for all Z for the case of expected disablement, or for more than one present symptom in the case of expected sufficiency or at all for expected disablement."*

Our comment on the case where $\tau(D_k, Z) \rightarrow 1$ is not intended to refer to a realistic case, but rather if we were to replace these values in equation (5), resulting in a new equation, this equation is precisely the posterior, which can be seen by decomposing the joint disease-symptom marginal in (5) by applying the inclusion-exclusion principle [6],

$$\begin{aligned}
P(\mathcal{S}_\pm, D_k = 1|\mathcal{R}) &= P(\mathcal{S}_- = 0, \mathcal{S}_+ = 1, D_k = 1|\mathcal{R}) \\
&= \sum_{\mathcal{Z} \subseteq \mathcal{S}_+} (-1)^{|\mathcal{Z}|} P(\mathcal{S}_- = 0, \mathcal{Z} = 0, D_k = 1|\mathcal{R})
\end{aligned}$$

which yields the an expression that can be identified as the numerator of equation (5) if it were the case that $\tau(D_k, Z) \rightarrow 1$. This is a minor observation, intended to show that the expected sufficiency and disablement can be written in a ‘similar form’ to the posterior, and is without further importance to the findings of the article. We have removed this sentence to avoid confusion.

Point 10. *“In the author’s (and others) terminology a disease is a sufficient cause for a symptom if no other disease is needed to produce the symptom. This allows for the possibility that the disease occurs but the symptom does not. There is another use of the term “sufficiency” that is common in discussions of causality which would not allow for the latter possibility (e.g in discussions of INUS conditions for causation). They authors should include a footnote to avoid this confusion. This is clear in the appendices but not the main body of the paper.”*

A footnote and citation have been added to the final paragraph of section 2.

Further to these changes, we have included changes to the article to comply with all requirements for publication in this journal, including

1. Inclusion of data availability section.
2. Inclusion of code availability section.

Bibliography

- [1] Badgeley, Marcus A., et al. "Deep learning predicts hip fracture using confounding patient and healthcare variables." *NPJ digital medicine* 2.1 (2019): 1-10.
- [2] Janizek, Joseph D., et al. "An Adversarial Approach for the Robust Classification of Pneumonia from Chest Radiographs." *arXiv preprint arXiv:2001.04051* (2020).
- [3] Gowal, Sven, et al. "Achieving Robustness in the Wild via Adversarial Mixing with Disentangled Representations." *arXiv preprint arXiv:1912.03192* (2019).
- [4] Besserve, Michel, Rémy Sun, and Bernhard Schölkopf. "Counterfactuals uncover the modular structure of deep generative models." *arXiv preprint arXiv:1812.03253* (2018).

- [5] Kocaoglu, Murat, et al. "CausalGAN: Learning causal implicit generative models with adversarial training." arXiv preprint arXiv:1709.02023 (2017).
- [6] Te Sun, H. "Multiple mutual informations and multiple interactions in frequency data." Information and control 46 (1980): 26-45.

Reviewers' Comments:

Reviewer #4:

Remarks to the Author:

The revisions that the authors have made satisfy all of the suggestions that I made in the previous review. I recommend its publication. (The version of the revised paper that I received did not display two figures in Appendix I; however it was clear what they were supposed to be from the text.)

We are grateful to hear that our manuscript has been accepted in principle, pending editorial corrections. Please find in this letter a point-by-point response to all reviewer comments.

Reviewer 4 (Remarks to the Author):

The revisions that the authors have made satisfy all of the suggestions that I made in the previous review. I recommend its publication. (The version of the revised paper that I received did not display two figures in Appendix I; however it was clear what they were supposed to be from the text.)

response: The reviewer suggestions no further suggestions and recommends the manuscript for publication.